# Nightbeat: Heart Rate Estimation From a Wrist-Worn Accelerometer During Sleep

Max Moebus[1]  Lars Hauptmann[1]  Nicolas Kopp[1]  Berken Demirel[1]  Björn Braun[1]  Christian Holz[1]

*Abstract*—Today's fitness bands and smartwatches typically track heart rates (HR) using optical sensors. Large behavioral studies such as the UK Biobank use activity trackers without such optical sensors and thus lack HR data, which could reveal valuable health trends for the wider population. In this paper, we present the first dataset of wrist-worn accelerometer recordings and electrocardiogram references in uncontrolled at-home settings to investigate the recent promise of IMU-only HR estimation via ballistocardiograms. Our recordings are from 42 patients during the night, totaling 310 hours. We also introduce a frequency-based method to extract HR via curve tracing from IMU recordings while rejecting motion artifacts. Using our dataset, we analyze existing baselines and show that our method achieves a mean absolute error of 0.88 bpm—76% better than previous approaches. Our results validate the potential of IMU-only HR estimation as a key indicator of cardiac activity in existing longitudinal studies to discover novel health insights. Our dataset, Nightbeat-DB, and our source code are available on GitHub: github.com/eth-siplab/Nightbeat.

*Index Terms*—Accelerometers, Heart rate, Sleep

## I. Introduction

**W**RIST-worn accelerometers are commonly used in longitudinal studies to analyze patients' activity patterns, such as exercise routines, physical activity metrics, or sleep and wake times. For example, the UK Biobank combines wrist-based accelerometer recordings from 100,000 patients during a week-long sub-study [1]. This study alone has revealed various activity and sleep patterns as risk factors for depression [2], cardiovascular disease [3], different types of cancer [4], and overall mortality [5]. Besides the accelerometer, no other sensing modality was included. Because the UK Biobank and comparable studies [6], [7] pair week-long accelerometer recordings with decades of electronic health records, using the accelerometer recordings to extract physiological features that are otherwise not included in the study could advance early disease detection and risk factor identification on a population level. Heart rate dynamics have been shown to link to sleep quality and the progression of rare diseases alike, hence, showing great value for identifying novel risk factors on a population level [8]–[12].

Recent studies have shown promise for analyzing cardiac activity from the data recorded by body-worn accelerometers, specifically heart rate (HR [13], [14]). These approaches leverage the ballistocardiogram (BCG [15], [16]), which captures the subtle mechanical vibrations caused by heartbeats.

Extracting robust HR estimates from BCG recordings in real-world settings is a substantial challenge, particularly when

sampled from the wrist. Even simple daytime activities such as walking or eating exhibit larger motion magnitudes than those in the BCG [13], [15], rendering such estimation near impossible in practical settings. Moments of sleep mark a notable exception—they lend themselves to sampling BCG signals, as voluntary movements are minimal [13]–[15]. Previous studies have extracted HR values from individual patients' recordings in controlled sleep laboratories [13]–[15], where sensor placement was verified by staff. Here, even individual heartbeats can be detected from wrist-worn accelerometers during sleep [13], as long as motion-afflicted segments are reliably removed (e.g., up to 80% of recordings [14]). Despite this promise, no existing methods reach practical accuracies in their estimates for uncontrolled real-world environments—and no datasets with continuous ground-truth references exist to facilitate and evaluate developments of the former.

In this paper, we introduce Nightbeat-DB, a novel dataset for BCG-based HR estimation tasks during sleep. Our dataset replicates the settings of large longitudinal studies such as the UK Biobank [1] to allow for the evaluation of techniques in similarly uncontrolled environments. 42 patients received a wrist-worn activity tracker (Axivity AX3, which embeds a 3-axis accelerometer) for wear on their dominant wrist for one night at home while following their regular sleep routine at home. Patients also received an ECG chest belt (movisens EcgMove 4) to record reference signals for ground-truth values of HR and inter-beat intervals (IBI). Patients also indicated if they shared their beds or slept alone, which is important for BCG analyses, as vibrations couple through bed mattresses and can lead to interfering signals [17].

We complement the dataset with Nightbeat, an estimation method for HR from 3-axis wrist-based accelerometer recordings by removing motion artifacts, tracing HR curves in the frequency domain, detecting individual heartbeats after a filter and frequency detection stage, followed by simple post-processing. We validate our method on Nightbeat-DB and show its robustness to intermittent motion during the night. Compared to 3 baseline methods, our method achieves the lowest error ($MAE = 0.88$ bpm), lowering the error of previous approaches ($MAE = 3.68$) by 76%, while removing the same amount of data (22%) [13]. For female participants, we find that sharing a bed leads to an average increase in MAE of 32%. For male participants, the difference is only 2%.

In summary, this paper makes the following contributions:

1. a novel dataset of continuous wrist-based accelerometer signals and paired ECG reference signals during the night for HR estimation tasks, designed to replicate the uncontrolled

[1]Department of Computer Science, ETH Zurich, Zürich, Switzerland
Emails: firstname.lastname@inf.ethz.ch (e.g., bjoern.braun@inf.ethz.ch)

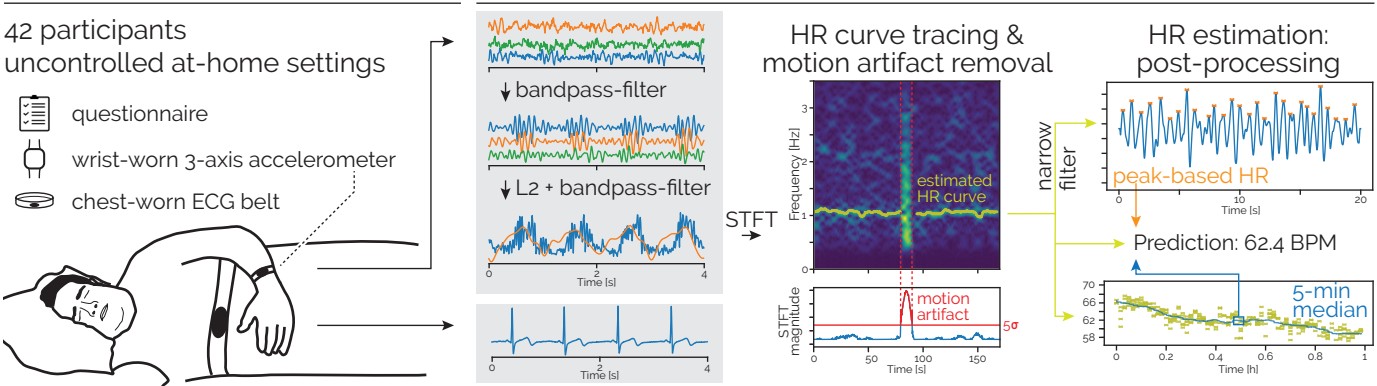

Fig. 1. Our dataset comprises continuous motion signals from a wrist-worn 3-axis accelerometer (Axivity AX3, 100 Hz) and corresponding ECG signals (movisens EcgMove 4) from 42 patients and nights. From the ballistocardiogram (BCG) captured by the vibrations reaching the wrist-based sensor, our signal processing method estimates the patient's heart rate using a combination of filters and heuristics for pre-processing, a short-term frequency analysis to identify HR curve, and a filter stage to precisely select the peak of the BCG wave for inter-beat interval detection. We combine the HR curve with the detected inter-beat intervals and a 5-minute median smooth to make a prediction for every 20-second window.

at-home setting of existing large-scale datasets that do not include continuous HR recordings "in-the-wild" [1], [6], [7],

2. a signal processing method for HR estimation from accelerometer signals that rejects motion artifacts, detects the HR via curve tracing in the frequency domain, and estimates heartbeats and inter-beat intervals, and

3. a comparison of existing approaches for HR estimation from wrist-worn accelerometers using BCG sensations on our dataset in the context of uncontrolled at-home settings where some patients share their beds. Our method achieves the lowest error compared to 3 (recent) baseline methods ($MAE = 0.88$, $76\%$ reduction compared to recent baselines) and an average correlation of $0.81$ across participants.

Taken together, we demonstrate the suitability of our method for application to existing and future datasets collected in free-living and at-home conditions with activity monitors that do not include alternative sensors for HR estimation (e.g., UK Biobank). Our method achieves accuracy that is suitable for medical analysis [13], making it valuable for future activity studies and monitoring cardiac activity in natural settings.

## II. RELATED WORK

### A. HR Estimation From Wearable Sensors

Most modern wearables use optical sensors for HR estimation. While highly successful in periods with little motion, HR estimation based on Photoplethysmography (PPG) becomes difficult in the presence of motion artifacts [15]. While a recent method fuses PPG and accelerometer signals in a probabilistic deep learning framework to deal with motion artifacts [18], most works still employ simple rules on aggregates of the underlying signal or the short-time Fourier transform (STFT) to identify segments corrupted by motion artifacts [19]. After reliably detecting corrupted segments, HR estimation on the remaining uncorrupted signals proves a much simpler task where well-designed signal processing methods often outperform (convolutional) neural networks [19].

### B. Signal Quality Estimation

Estimating the quality of a signal segment (e.g. detecting motion artifacts) has been crucial in health monitoring across various sensing modalities [20]–[22]. Most of these methods extract several features, so-called signal quality indices (SQI), from the signals and employ rule-based decision algorithms to detect segments with artifacts [21]. Alternatively, agreement in prediction between different sensors or template matching have been used to identify segments of high signal quality [21], [22]. Although some learning-based algorithms have been proposed [20], they require ground truth labels, which must be annotated manually by an expert. In this work, we propose a method for motion artifact detection based on the energy in the STFT that does not require ground truth annotations for training. We further use disagreement between different estimation techniques to identify potentially corrupted segments.

### C. Datasets for Accelerometer-Based HR Estimation During Sleep

Accelerometers have a decades-long history in sleep science [23], yet only few datasets are suitable for HR estimation from wrist-worn accelerometers during sleep. Besides sampling at a sufficient rate, accelerometers must also be sensitive enough to detect the subtle BCG waveforms. Currently available datasets that allow to detect BCG waveforms, however, have been recorded in sleep laboratories, where signal quality is higher than in uncontrolled at-home settings.

Multiple datasets are suitable for our task of HR estimation from wrist-worn accelerometers [13], [15], [24]. Walch et al. recorded a dataset with 32 participants for sleep staging based on signals supplied by the Apple Watch, i.e. a 3-axis accelerometer sampling at 50 Hz and HR values supplied by the Apple Watch's PPG sensor every 5–15 seconds [24], [25]. For the task of sleep staging, the dataset was recorded in a sleep laboratory. Also inside a sleep laboratory, Zschocke et al. recorded a dataset using the SOMNO-Watch comprising

of more than 200 participants to reconstruct respiration and HR signals [14], [15]. Similarly, Weaver et al. collected data from 82 children inside a sleep laboratory and compared the performance of an Apple Watch and ActiGraph GT9X for accelerometer-based HR estimation [13].

Current datasets have been recorded in sleep laboratories [13], [15], [24], which limits their use for the purpose of HR estimation from wrist-worn accelerometers in uncontrolled at-home settings. Sleep laboratories are not representative of at-home settings, mainly since sleep laboratories are limited to participants sleeping alone in their beds and thus omit a potentially crucial source of noise. Motions emitted during the cardiac cycle are measurable through a mattress [17] leading to potential interference if two people share a bed. Furthermore, in large studies such as the UK Biobank, participants receive the accelerometer via mail [1] and are instructed to put it on themselves. In sleep laboratories, trained personnel verify that sensors are worn correctly and ensure signal quality.

### D. Accelerometer-based HR Estimation During Sleep

Accelerometers have been used in diverse settings for HR monitoring, even though not currently when worn on the wrist in completely uncontrolled settings. Seismocardiography, when an accelerometer is placed on the chest, is likely the most common setting [26]. Without motion artifacts, it is then possible to observe the characteristic BCG waveform [27].

The characteristic BCG components are also visible if an accelerometer is integrated into a chair [28] or placed under a scale [29] that the participant is standing or sitting on. During sleep, accelerometers have been successful at HR as well as breathing rate estimation if placed on or under a mattress [17], [30]. For wrist-worn accelerometers, multiple approaches have been proposed for extracting HR and heart-beats during sleep [13], [15], [16]. For example, pulse wave reconstruction (PWR) [14], [15] applies a bandpass filter with cut-off frequencies of 5–14 Hz followed by a Hilbert transform to detect peaks on the axis with the highest autocorrelation in the plausible range of HR values. The authors also exclude segments where the total magnitude is above 5 mg and segment the accelerometer signal based on ECG peaks to demonstrate correlations of inter-beat intervals of up to 0.91. On average they ignore everything apart from 1.7 h per night per participant. The requirement for ground truth ECG peaks makes the approach impractical "in-the-wild". Similarly, BioInsights [16] requires completely movement-free periods for HR extraction from wrist-worn accelerometers without presenting a technique to remove motion artifacts. Even during low-movement periods such as asleep, this is impractical. As a result, both PWA and BioInsights highlight the need for more robust methods that can accurately extract HR data despite the presence of motion artifacts. A recent study by Weaver et al. investigated extracting HR from accelerometers in more realistic sleep settings for children [13]. The authors proposed calculating the derivative of the total magnitude of the wrist-worn accelerometer, followed by root mean square averaging, and then applying the Hilbert transform while using

a short-time Fourier transform for HR detection. Using the accelerometer signal of an Apple Watch (sampled at 50 Hz), they achieve a mean absolute error of 6.4 bpm, which increases to 16.8 bpm when using an ActiGraph GT9X (sampled at 100 Hz). The authors use the rolling standard deviation of the STFT peak-width (140 seconds) to estimate signal quality.

In this work, we present Nightbeat-DB, a comprehensive dataset recorded in at-home settings, for a more robust evaluation of HR extraction methods. Further, we present Nightbeat, a HR estimation method capable of detecting motion artifacts and accurately tracking the HR across the entire night.

## III. NIGHTBEAT-DB DATASET

To study the performance of HR estimation tasks based on wrist-worn accelerometers during sleep outside sleep laboratories, we recorded a dataset with 42 patients "in the wild." In contrast to existing datasets, the characteristics of our dataset mimic the setting of existing large-scale health studies that include wrist-worn accelerometers, yet no HR signals.

### A. Study Setup and Participants

All participants received a 3-axis accelerometer for wear on their dominant wrist (Axivity AX3) and an ECG chest belt (movisens EcgMove 4). Participants received the same simple and short instructions as those in the UK Biobank motion substudy [1] to ensure a little-controlled experimental setting. Participants were instructed to wear the accelerometer such that the z-axis was orthogonal to the skin surface and that the wristband did not 'jiggle' yet was still comfortable for a single night: from shortly before going to bed until shortly after waking up. Similarly, participants were instructed to tightly wear the ECG belt, but not to tighten it uncomfortably. Further, participants were asked to fill in a questionnaire after waking up about demographic information, the time they went to bed and woke up, and if they slept alone. Participants who did not sleep alone indicated if they slept on a separate mattresses.

All 42 participants provided written informed consent and the study protocol was approved by the ethics committee of ETH Zurich: 23 ETHICS-002. 42 participants were recruited via convenience sampling in and around Zurich (18 female, 24 male, ages 20–36, M=26.48 years). None reported to suffer from a sleep disorder. 15 participants reported that they shared their bed with someone else when participating in the study. None of these 15 participants slept on a separate mattress.

### B. Recorded Signals and Sensor Synchonization

The Axivity AX3 recorded motion signals at the dominant wrist at 100 Hz (± 8 g). The movisens EcgMove 4 recorded a 1-channel ECG signal at 1024 Hz. The movisens EcgMove 4 further recorded motion from the chest at 64 Hz (± 16 g), which we used to synchronize the two sensors. Before participants were handed the two devices, we initiated the recording and included a synchronization pattern by laying the two devices on top of one another and hitting them against a book several times. After we received back the two devices, we included another pattern for synchronization

and stopped the recording. To synchronize the signals from the two devices, we upsampled the motion signal from the movisens EcgMove 4, manually spotted the approximate time stamps of the synchronization patterns, and then used the cross-correlation between the two motion signals to accurately align them (motivated by the procedure described in [31]).

### C. Ground-Truth HR Measurements from ECG

We extracted reference HR values as ground truth from ECG recordings in two ways. First, we used the vendor's built-in algorithm to obtain continuous HR measurements (movisens EcgMove 4). Second, we additionally processed the ECG recordings with a series of methods implemented as part of the NeuroKit2 Python library [32], including popular algorithms such as Pan-Tompkins [33] but also a more modern wavelet-based approach [34]. We compared the output from all algorithms with that of the vendor's implementation and manually verified each signal portion for disagreements in HR estimates. Since no inspected signal recordings were of low quality, we selected HR estimates from the vendor. For a detailed inspection, please visit our GitHub repository.

### D. Data Preparation & Format

We made use of the approximate sleep and wake times provided by participants to set the start and end points for our processing pipeline. Subsequently, we used the provided signal quality metric by the movisens EcgMove 4 (provided at $1/60$ Hz) to identify the first 60-second window after the sleep time provided by participants when the participant wore the movisens EcgMove 4. From there until the last 60-second window when participants wore the movisens EcgMove 4 before their approximate wake time, we generally processed the synchronized raw data in 3-minute windows with a 2-minute overlap. 3-minute processing windows proofed long enough such that our method of motion artifact detection and HR curve tracing worked well (Section IV-B and IV-C). For the central 60-second window, we primarily relied on the data quality metric provided by the movisens EcgMove 4 to detect potential non-wear, ensure the participants wore the ECG belt correctly such that a meaningful ECG signal was measurable, and that there were no motion artifacts that affected at least 5 consecutive heartbeats. As part of our proposed approach, we removed further motion artifacts as outlined in Section IV-B.

## IV. NIGHTBEAT HR ESTIMATION METHOD

Our proposed method estimates HR from BCG observations in the accelerometer signal based on the L2-norm of the three bandpass filtered axes of the accelerometer. We will refer to this transformation as the Nightbeat signal (Section IV-A). After computing the Nightbeat signal, our algorithm consists of four main parts: 1: Section IV-B) removal of motion artifacts, 2: Section IV-C) peak tracing in the frequency domain to detect the most likely HR trajectory, 3: Section IV-D) peak detection in the time domain to identify heartbeats, and 4: Section IV-E) simple post-processing across 5-minute windows. Signals are split into 3-minute windows with a 2-minute overlap to identify motion artifacts and to detect the overall trajectory of participants' HR by tracking the peaks in the STFT using a method inspired by curve tracing for PPG [19]. We then split the sections free of motion artifacts into 20-second windows with a 10-second overlap and take the mean value of the estimated HR trajectory for this window as our frequency-based HR prediction ($HR_C$). We then narrowly filter the Nightbeat signal based on the frequency-based HR prediction to detect heartbeats via peak detection in the time domain and derive a HR prediction based on inter-beat intervals $HR_P$. Finally, we compute the rolling median of the frequency-based HR predictions across 5-minute windows ($HR_M$). If $|HR_C - HR_P| \leq 10$ and $|HR_C - HR_E| \leq 10$, we predict $HR_C$ for a window, else we do not make a prediction.

### A. Nightbeat Signal: Filtering & L2-norm

We filter each of the three axes using a $4^{th}$-order Butterworth bandpass filter to isolate frequencies in the range of 5–14 Hz and compute the L2-norm of the 3 filtered axes. We then band-pass filter it using a $4^{th}$-order Butterworth filter to the range of 0.5 to 3.5 Hz to arrive at what we refer to as the Nightbeat signal. The procedure is displayed in Figure 1.

### B. Artifact Removal

Even though abrupt movements occur less frequently during sleep than while awake, the accelerometer signal is not free of such motions. We detect motion artifacts directly from the time-frequency representation of the 3-minute Nightbeat signal. To do so, we compute the STFT with a window length of 1024 samples (Kaiser window function with $\beta = 2$) padding each window by factor 10 and shifting it by 10 samples. This resulted in resolutions of 0.2 seconds and 0.01 Hz, in the time and frequency dimensions, respectively.

From the STFT of the 3-minute window, we sum the magnitude across all frequency bins for each timestep and thus derive a distribution of total spectral energy over time. As shown in Figure 2, we identify segments as motion artifacts where the total spectral energy lies 5 standard deviations ($\sigma$) above the 3-minute median. 3-minute windows proofed long enough such that motion artifacts differ significantly from the median. We estimate $\sigma$ robustly using the interquartile range (IQR; the IQR of $N(\mu, \sigma)$ is $1.349\sigma$).

### C. Frequency Peak Identification: Curve Tracing

Based on the same STFT output used for artifact removal in Section IV-B (i.e., STFT of the 3-minute Nightbeat signal), we estimate HR values via curve tracing during periods free of motion artifacts [19]. While our method is inspired by [19], we do not use gradients to identify curves but simply track possible peaks in the frequency domain across consecutive timesteps—allowing for gaps and jumps.

At each timestep (every 0.2 seconds), we identify all peaks in the frequency domain (0.5–3.5 Hz) whose height and prominence exceed half of the highest magnitude in the range of 0.5–3.5 Hz. As visible in Figure 2, BCG harmonics are often visible at multiples of the HR. We ignore any peaks that are

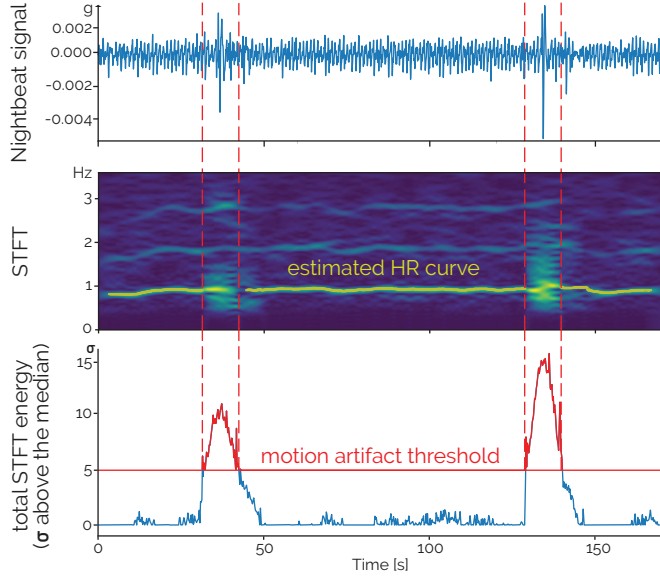

Fig. 2. We detect motion artifacts during 3-minute windows based on the energy of the STFT of the Nightbeat signal over time. If the energy lies 5 standard deviations ($\sigma$; estimated robustly via the IQR) above the median, we flag the segment as a motion artifact.

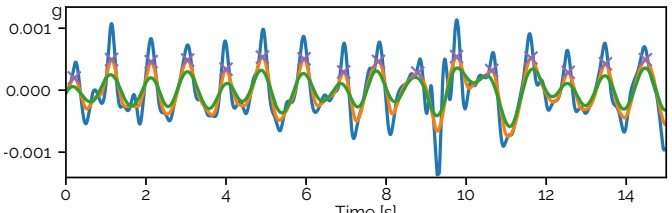

Fig. 3. Peak detection (purple crosses) from the Nightbeat signal (blue) for an exemplary 15-second signal. The smoothed and filtered signal is displayed in orange, the height threshold for peak detection in green.

a multiple of an identified lower peak (up to a deviation of 2%) in the frequency domain. After identifying all valid peaks at every timestep, we connect neighbouring peaks to what we refer to as 'HR curves'. Starting at the first timestep, all identified peaks in the frequency domain start a new HR curve. Moving to the next timestep, all identified peaks are either attributed to an existing HR curve if they lie within 0.05 Hz of a peak at the previous timestamp or start a new HR curve. If for any identified peak there was no peak at the previous timestep within 0.05 Hz we looked up to 5 timesteps back—effectively allowing for 1 second gaps in HR curves. We then remove all curves that last less than 10 seconds. We are then left with potentially multiple curves at the same timestep. We proceed by calculating the median frequency of all points attributed to any curve that is longer than 10 seconds and their IQR. We then ignore any curve whose median is further than 5 $\sigma$ away from the overall median—where the $\sigma$ is again robustly estimated via the IQR. In the case that there are two curves remaining for the same timestep, we select the curve closest to the overall median. We then arrive at curves such as the ones displayed in Figure 1 and Figure 2, where there is at most one HR curve per timestep but the curves might have gaps. We interrupt any HR curves during motion artifacts and start anew after the identified motion artifact.

For each 20-second window, we then arrive at a frequency-based HR prediction based on the mean of the detected HR curve during this window: $HR_C$ (in Hz). We skip any window where no HR curve has been detected.

### D. Heart Beat Detection

Based on $HR_C$ (in Hz) for a 20-second window, we perform peak detection in the time domain on a signal narrowly filtered around $HR_C$. First, we apply the rolling median across $0.3\,\mathrm{s}/HR_C$, followed by taking the rolling mean across the same window length to lowpass filter and smooth the signal. We then narrowly bandpass filter this signal to the range of $0.9 HR_C$–$1.1 HR_C$ using a $20^{th}$-order FIR filter. We then detect all peaks with a height exceeding a rolling mean across $0.5\,\mathrm{s}/HR_C$ (Figure 3). We then average the inter-beat intervals to arrive at our second HR prediction: $HR_P$. If $|HR_C - HR_P| > 10\,\mathrm{bpm}$, we do not make a prediction. In our GitHub repository, we further compare alternative approaches.

### E. Post-Processing: Rolling Median

Our post-processing uses the rolling median of $HR_C$ (after removing windows where $|HR_C - HR_P| > 10\,\mathrm{bpm}$) across 5-minute windows: $HR_M$. If $|HR_C - HR_M| \leq 10\,\mathrm{bpm}$, we predict $HR_C$, else no prediction is made. We found the rolling median to perform well due to its robustness. Similarly to heart beat detection, the rolling median serves as a quality check and must primarily reach a satisfactory level.

### V. VALIDATION

We evaluate Nightbeat on Nightbeat-DB and the AW dataset (Section V-A) against 3 baselines (Section V-B). Together with Nightbeat-DB, and our Nightbeat method, all relevant code to replicate our analysis is available via GitHub.

### A. AW Dataset

In addition to Nightbeat-DB, we used a dataset recorded with an Apple Watch (Series 2 & 3) that comprises 31 participants (21 female) where motion data from a 3-axis accelerometer was recorded at 50 Hz from the wrist [24], [35]. HR values are available approximately every 5 seconds based on the Apple Watch's PPG sensor. Initially, the dataset was recorded to assess sleep staging algorithms. Participants were aged 19—55 years (29.4 years on average) and the available data was recorded during the participants' stay in a sleep laboratory. Only HR estimates supplied by the Apple Watch are available. We will refer to this dataset simply as the 'AW' dataset. It is publicly available via Physionet [25].

We processed the AW dataset similarly to our Nightbeat-DB dataset (Section III-D) based on 3-minute windows with a 2-minute overlap. However, we used different data requirements adapted to the AW dataset. The Apple Watch returned slightly irregularly sampled acceleration signals. Due to the

TABLE I
COMPARISON OF DIFFERENT ALGORITHMS FOR HR PREDICTION FROM
WRIST-WORN ACCELEROMETERS WHILE ASLEEP.

| approach | dataset | MAE | RMSE | Cor |
|---|---|---|---|---|
| BioInsights [16] | Nightbeat-DB | 17.47 | 21.38 | 0.00 |
| | AW | 21.12 | 24.53 | -0.03 |
| Jerks [13] | Nightbeat-DB | 13.23 | 20.63 | 0.04 |
| | AW | 3.68 | 7.11 | 0.50 |
| PWR [14] | Nightbeat-DB | 8.83 | 10.91 | 0.22 |
| | AW | 6.05 | 7.77 | 0.37 |
| Base: Mean-Pred. | Nightbeat-DB | 3.16 | 4.07 | 0 |
| | AW | 2.98 | 4.09 | 0 |
| **Nightbeat (ours)** | Nightbeat-DB | **0.88** | **2.24** | **0.81** |
| | AW | **1.68** | **3.38** | **0.64** |

TABLE II
PERFORMANCE OF NIGHTBEAT DEPENDING ON SEX AND WHETHER
PARTICIPANTS SLEPT ALONE (NIGHTBEAT-DB).

| approach | sex | alone | MAE | RMSE | Cor |
|---|---|---|---|---|---|
| Nightbeat (ours) | F | Y | 0.78 | 1.89 | 0.87 |
| | F | N | 1.03 | 2.30 | 0.80 |
| | M | Y | 0.86 | 2.49 | 0.77 |
| | M | N | 0.88 | 2.05 | 0.84 |
| | All | Y | 0.83 | 2.23 | 0.82 |
| | All | N | 0.94 | 2.15 | 0.82 |
| Base: Mean-Pred. | F | Y | 3.01 | 3.95 | 0 |
| | F | N | 2.71 | 3.47 | 0 |
| | M | Y | 3.19 | 4.28 | 0 |
| | M | N | 3.01 | 3.86 | 0 |
| | All | Y | 3.11 | 4.14 | 0 |
| | All | N | 2.88 | 3.70 | 0 |

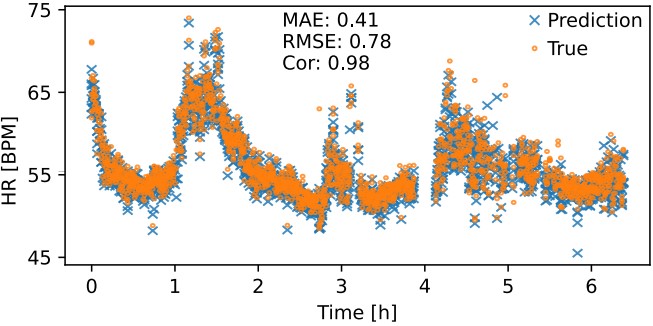

Fig. 4. Comparison of Prediction and ground-truth HR over time for P11 of the Nightbeat-DB dataset. Using our Nightbeat HR estimation method, we achieve an MAE of 0.41 and a correlation of 0.98.

specifications of our algorithm as outlined in Section IV, we required that no gap between consecutive samples exceeded 0.05 seconds and that no gap between consecutive HR values exceeded 10 seconds. If these requirements were met, we upsampled the output of the accelerometer signal to 100 Hz via linear interpolation to make the sampling rate match our Nightbeat-DB dataset. In total, the data amounted to 113 hours.

### B. Implementations of Alternative Approaches

In addition to Nightbeat outlined in Sections IV-A–IV-E, we compare three approaches from related works described in Section II-D: BioInsights by Hernandez et al. [16], Pulse Wave Reconstruction (PWR) by Zschocke et al. [14], [15] and Jerks are Useful (Jerks) by Weaver et al. [13]. To the best of our knowledge, these three approaches summarize the current works for HR estimation from wrist-worn accelerometers while asleep. If applicable, we implemented the signal quality metrics proposed by the respective approach and removed the same percentage of 20-second windows according to this metric as removed by Nightbeat (approx. 22%) to allow for fair comparison. However, Zschocke et al. also remove beats during their evaluation if they are not detected within a specified window following an ECG peak [14]. Since we want the methods to be completely independent of any signal not derived from the accelerometer, we did not implement this last quality check. We reimplemented all approaches in Python and they can be found in our Github repository.

Besides the three approaches described above, we implemented a personalized Mean-Baseline (Base: Mean-Pred) that simply predicts the average HR label for each participant.

### VI. RESULTS

Table I shows the results of Nightbeat against three baseline approaches. We use the mean absolute error (MAE), root mean squared error (RMSE), and correlation (Cor) as metrics to compare the different approaches across the two datasets.

Nightbeat outperforms all Baselines and achieves an MAE of as low as 0.88 bpm averaged across all participants on Nightbeat-DB. Averaged across individual participants, we achieve a correlation of 0.81. For a single participant, we achieve a correlation as high as 0.98 on Nightbeat-DB at an MAE of 0.41 bpm (Figure 4). Across all participants, the correlation exceeds 0.95 (Figure 5). Out of all possible 20-second windows, we remove 22% due to estimated motion artifacts, gaps in HR curves, or due to the $HR_C$ prediction lying further than 10 bpm away from either of the $HR_P$ or $HR_M$ prediction. On the AW dataset, we achieve an MAE of 1.68 bpm at an average correlation of 0.64. Nightbeat is the only method that achieves lower errors than when predicting an individual's average HR during the entire night.

PWR and Jerks achieve MAEs of 8.83 and 13.23, respectively on the Nightbeat-DB. On the AW dataset, PWR and Jerks achieve lower MAEs of 6.05 and 3.68 bpm, respectively, at average correlations of 0.37 and 0.50. The BioInsights approach achieves high MAEs of around 20 bpm on both datasets at a correlation of approximately 0. None of the three approaches achieves MAEs below 3.36 or 2.98 bpm, which is the performance of predicting each participant's average HR on Nightbeat-DB and the AW dataset (Base: Mean-Pred.).

In Table II, we notice a 32% increase in MAE for female participants when sharing a bed (based on Nightbeat). For male participants, the MAE does not increase when sharing a bed. In fact, it decreases by 2%. On Nightbeat-DB, MAE was not significantly correlated with age ($r = 0.18, p = 0.27$), weight ($r = 0.03, p = 0.84$), size ($r = -0.05, p = 0.77$), and average accelerometer magnitude ($r = -0.20, p = 0.24$).

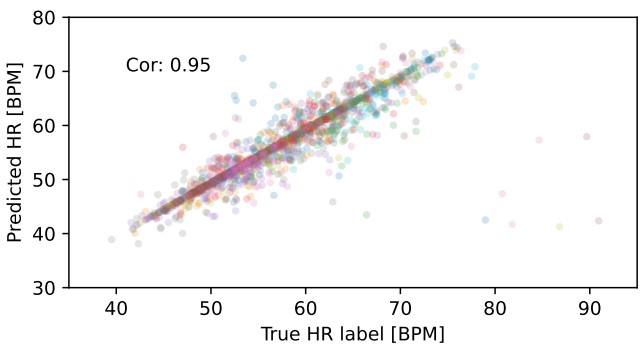

Fig. 5. Predictions against ground-truth HR for all participants of the Nightbeat-DB dataset—colored by participants. Across all participants, our Nightbeat Hr estimation method achieves a correlation of 0.94.

## VII. Discussion

With Nightbeat and Nightbeat-DB, we present a novel dataset that closely replicates the setting of large longitudinal studies accompanied by a novel approach that is suitable for HR prediction in such uncontrolled environments. Various large longitudinal studies comprise week-long accelerometer recordings and decades of health records that are constantly being updated, yet lack continuous HR signals "in-the-wild". Using continuous HR signals (while asleep), we might advance our understanding of diseases such as cardiovascular disease, mental disorders, or even neurological diseases and their respective disease progression and risk factors. Compared to (recent) baselines, Nightbeat is the only approach that exceeds mean prediction in terms of MAE during the night and reduces errors by at least 76% across datasets. We thus take a large step towards HR estimation while asleep in uncontrolled studies such as the UK Biobank accelerometer study that currently lack HR signals for medical analysis.

Nightbeat performs almost twice as well on Nightbeat-DB, despite Nightbeat-DB posing a greater challenge given its less controlled environment compared to the AW dataset. The AW dataset comprises HR estimates derived from the Apple Watch's PPG sensor every 5–15 seconds. To derive ground-truth HR values for the AW dataset, we essentially linearly interpolate the labels provided by the Apple Watch. Likely, this leads to inaccuracies in the labels and an effective glass ceiling for performance. Even though HR varies less during sleep than while awake, HR readings from up to 10 seconds ago, will still likely differ from momentary assessments.

In contrast to Nightbeat, PWR and Jerks perform better on the AW dataset. This might be indicative of the more challenging environment of at-home settings compared to sleep laboratories. Given that we observe a 32% increase in MAE for female participants when sharing a bed on Nightbeat-DB, the AW dataset should pose an easier task. However, the increase in MAE from 3.68 to 13.23 for Jerks is surprising. In their original manuscript, Weaver et al. evaluated Jerks on a dataset recorded using an Apple Watch and an Actigraph GT9X [13]. Interestingly, they observed a similar pattern and reported an MAE of 6.4 when using the Apple Watch, but an MAE of 16.8 when using the GT9X. This might indicate that their jerk-based approach is sensitive to the accelerometer in use. As arguably the simplest baseline and without motion artifact removal, it is unsurprising that Bioinsights does not cope well in realistic scenarios. While significantly more sophisticated than BioInsights, PWR seems optimized for segments with very high signal quality, which might explain its relatively low performance if only removing 22% of recordings.

Nightbeat combines a relatively simple, yet robust signal aggregation, with a sophisticated approach for HR estimation that makes use of the relatively slow-moving HR dynamics while asleep. However, our methods for motion artifact removal, HR curve tracking, and post-processing are unlikely to extend well to scenarios where participants are awake. While awake, movements occur much more frequently and for much longer time periods, which our method of motion artifact removal might struggle with. Similarly, our method of curve tracing and post-processing will likely struggle with the few suitable windows for HR prediction and potentially much faster changing HR values.

## VIII. Conclusion

We demonstrated that during sleep wrist-worn accelerometers provide a robust alternative for HR estimation—also in uncontrolled at-home settings where people might share their bed. In accelerometer studies that do not include HR signals, this introduces new opportunities for analysis. For accelerometer studies that were integrated into large longitudinal efforts such as the UK Biobank such methods will be of particular value for early disease detection and to enhance our understanding of disease progression and risk factors.

Future work might evaluate the value of Nightbeat for disease modeling (i.e., apply Nightbeat to accelerometer studies of large longitudinal efforts) and extend similar methods to motion-free periods while participants are awake. To extend heart rate estimation techniques to uncontrolled settings while awake, the identification of suitable periods and robust motion artifact removal will likely be crucial. Estimated HR trajectories from such methods might then be used to further understand risk factors of diseases including cardiovascular disease, neurological conditions, or mental disorders.

## IX. Limitations

Nightbeat-DB, Nightbeat, and our validation has limitations.

First, on both Nightbeat-DB and the AW dataset, we exclude segments when there are no ground-truth HR labels available. This makes the task of HR estimation on the remaining segments seem slightly easier since both ECG and PPG are affected by motion artifacts too. If no HR ground-truth could be derived, this is likely due to strong motion artifacts. This is a general problem of in-the-wild evaluations.

Further, on both datasets, HR labels are relatively low (usually below 90 bpm; see Figure 5). However, cardiovascular diseases can lead to different HR patterns during the night and the two datasets are thus not fully representative. While

Nightbeat does not constraint HR predictions to low ranges, in contrast to the original Bioinsights implementation with filter settings of 0.75–2 Hz, we lack datasets to adequately evaluate approaches for patients with higher HR while asleep.

One drawback of our approach is that it cannot track large HR changes that occur abruptly ($> 10$ bpm within a few minutes). Given our median smoother during post-processing, such abrupt changes may be tracked by the HR curve and detected in the time domain, but they might be rejected based on the fitted median smooth. While we did not observe such behavior on the two datasets used for validation, it would need evaluation on patients with sleep disorders and cardiovascular diseases to verify Nightbeat's accuracy for such patients.

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
