# OpenReview forum: "Nightbeat: Heart Rate Estimation From a Wrist-Worn Accelerometer During Sleep"
_IEEE.org/EMBS/BHI/2024/Conference — IEEE BHI'24_

### Official Review · Reviewer_wTKE · 2024-08-04
**Review: Nightbeat: Heart Rate Estimation From a Wrist-Worn Accelerometer During Sleep**

**Overall Rating:** 6
**Confidence:** 4

**Other Quality Metrics:**

- Clarity of writing: **great**
- Clinical Significance: **great**
- Methodological Novelty: **good**
- Experiments and Results: **great**

**Questions For The Authors:**

- What does "15 participants reported that they shared their bed with somebody else—none on a separate mattress" mean?

**Strengths:**

- **Clinical Significance**: The manuscript presented clear motivation and connected to a clinically significant problem.
- **Study design**: The manuscript presented a novel dataset with a detailed protocol design.
- **Methodology**: The manuscript presented clear details of the preprocessing pipeline.

**Summary Of The Paper:**

While remote and continuous HR estimation with body-worn IMU sensors raised attention, previous studies are only validated in controlled settings due to the lack of a labeled dataset. Therefore, the manuscript introduced the Nightbeat-DB dataset as a large longitudinal study for HR estimation in an uncontrolled setting. It also developed a processing method and comparisons with existing approaches to validate the dataset.

**Weaknesses:**

- **Methodology**: while the manuscript's explanation on signal processing was clear and compared with baselines, it lacks details on the HR estimation part.
- **Related Works**: the manuscript didn't explained what was the limitation of each previous datasets and how to that leads to the development of Nightbeat-DB.

---

> ### Author Rebuttal · Authors · 2024-09-01
>
> 1. **Details on the HR estimation part**\
> We have clarified Section IV further for our revision. Summary of our method:
>
> We compute the STFT of the Nightbeat signal over a 3-minute window. This is followed by four steps to ultimately predict HR values for 20-second windows.
>
> First, we identify motion artifacts via outlier detection from the magnitude of the STFT over time (Section IV.B).
>
> Second, we identify potential "HR curves" essentially by peak tracing in the STFT over time (Section IV.C). We refer to our HR estimate based on the HR curve as $HR_C$.
>
> Third, we filter the Nightbeat signal narrowly around $HR_C$ and perform peak detection to identify potential heartbeats (Section IV.D). We estimate the heart rate based on the detected inter-beat intervals: $HR_P$.
>
> Fourth, across all $HR_C$ where $\vert HR_C - HR_P \vert < 10 bpm$, we compute a rolling median: $HR_M$ (Section IV.E).
>
> Ultimately, we predict $HR_C$ only if $\vert HR_C - HR_P \vert < 10 bpm$ and $\vert HR_C - HR_M \vert < 10 bpm$. We don’t make a prediction 22% of the time.
>
> Effectively, $HR_M$ and $HR_P$ are quality checks.
>
> 2. **Limitations of previous datasets**\
> Previous datasets were recorded *inside* sleep laboratories and participants slept *alone* in a bed thereby ensuring higher signal quality and observing fewer motion artifacts than in uncontrolled settings.
> Nightbeat-DB is the first dataset for continuous HR estimation from a wrist-worn accelerometer that recorded participants in uncontrolled at-home settings, where some participants also shared their beds. Nightbeat-DB thus enables developing algorithms for existing large medical databases such as the UK Biobank.
> In the revision of our manuscript, Sections II.C and III.A are updated to mention the advantages of Nightbeat-DB more clearly right at the beginning.
>
> 3. **15 participants reported that they shared their bed with somebody else—none on a separate mattress**\
> We clarify: 15 participants reported that somebody else was also sleeping in the same bed during their recording. This *can* mean that the signal is corrupted due to additional motion artifacts caused by external events (see reviewer 5kAX reply 1).
>
> To account for two single mattresses in a double bed (e.g., two 80cm mattresses in a 160cm bed), which might have less of an effect, we had included a subquestion in our questionnaire. However, none of the participants who shared their bed slept on their own mattress.
> Our revision now clarifies the respective sentence.

---

### Official Review · Reviewer_Cfs1 · 2024-08-10
**Review: Nightbeat, Heart Rate Estimation From a Wrist-Worn Accelerometer During Sleep**

**Overall Rating:** 7
**Confidence:** 3

**Other Quality Metrics:**

- Clarity of writing: ***excellent***
- Clinical Significance: ***great*** (great dataset provided)
- Methodological Novelty: ***great*** (STFT based method is widely used)
- Experiments and Results: ***excellent***

**Questions For The Authors:**

N/A

**Strengths:**

- Great visualization of pipeline illustration for Nightbeat-DB to HR estimation flow and clear description of methodology as well as framework design
- Comprehensive review on previous works completed in sleep laboratory
- Novelty and significance of uncontrolled at-home setting wrist based accelerometer signals is clearly identified.
- Multiple methods are used for ground truth HR measurement from ECG signal to ensure data quality.
- Significant improvements are presented for HR prediction over baselines on MAE and RMSE.
- Limitation, especially constraints on rolling median smoothing is clearly stated.

**Summary Of The Paper:**

Introduce the first dataset of wrist-worn accelerometer recordings and electrocardiogram references in uncontrolled at-home settings to investigate the recent promise of IMU-only HR estimation via ballistocardiograms.

The recordings presented in this manuscript are from 42 patients during the night, totaling 310 hours. A frequency-based method is also introduced to extract HR via curve tracing from IMU recordings while rejecting motion artifacts. Existing baselines are examined and compared against proposed method, showing that proposed method, Nightbeat, achieves a mean absolute error of 0.88 bpm—76% better than previous approaches.

**Weaknesses:**

- In section III.C, it would be great to see a comparison of ground truth HR measurement from different processing methods.
- Demographics such as pre-existing health conditions, BMI, and lifestyle factors (e.g. alcohol cunsumption) are important factors that impede the framework's reliability of HR estimation. However, the manuscript's study cohort was relatively homogenous, with only healthy young adults under 40 years old.
- In section V, it would be great to see a comparison against alternative approaches on heart beat detection methods and a further discussion on the reason of superiority of rolling median method this manuscript proposed.

---

> ### Author Rebuttal · Authors · 2024-09-01
>
> **1. Ground truth HR measurement from different processing methods**\
> The differences between ECG peak detection algorithms are best analyzed using extensive plots since ultimately there exists no ground-truth for the detected peaks. We compared signals and detected peaks of different algorithms across clean, artifact-corrupted, and noisy signals.
> We believe such a comparison is too extensive to be included in the final manuscript. However, for fellow researchers, we will include a detailed comparison as part of our GitHub repository and refer interested readers to our GitHub repository in Section III.C of our final manuscript.
>
> **2. Relatively homogenous study cohort**\
> This might be our project's main limitation for practical applicability. We tried to stress this in our limitations section already (IX) and per the ethics approval will not be able to recruit a much more diverse participant group for now. However, we will stress this evaluation for future work as we pursue our project.
>
> **3. Alternative approaches for heart beat detection**\
> We estimate HR values primarily via the HR curves we detect from the STFT output. We only use the beat detection method as a first quality check to confirm that the HR estimated via the detected beats aligns reasonably well with the prediction based on the HR curve. Inaccurate beat detection methods would mainly increase the number of windows for which we do not make a prediction.
> We did compare other basic strategies for peak detection including for instance a wavelet-based approach. Ultimately, our choice was driven by the fact that in a clean signal, peaks are quite clearly visible to the human eye as can be seen in Figure 3. We found adaptive smoothing of the signal and a further smoothed height threshold work reliably even for difficult segments such as 8s to 12s in Figure 3.
> Given the strict page limit of 8 pages, we would mainly expand on this in our accompanying GitHub repository apart from relatively few changes to Section III.D.
>
> **4. Superiority of rolling median**\
> We evaluated the rolling median against a rolling mean and found its higher robustness to filter out erroneous predictions much better.
> Since we use it only to filter out predictions that deviate too much from the median trend, we found the simple median smoother to perform satisfactorily. We reckon that a more complex filter would only add minor improvements if any.
> Section IV.E of our revision now elaborates on this more.

---

### Official Review · Reviewer_5kaX · 2024-08-11
**Nightbeat: Heart Rate Estimation From a Wrist-Worn Accelerometer During Sleep**

**Overall Rating:** 7
**Confidence:** 4

**Other Quality Metrics:**

(a) Clarity of writing: Good
(b) Clinical Significance: Great
(c) Methodological Novelty: Good
(d) Experiments and Results: Good

**Questions For The Authors:**

I believe this manuscript represents an excellent study with significant relevance to "in-the-wild" research. Expanding on some of the details I mentioned regarding the analysis and results could further enhance the manuscript and strengthen its overall impact.

**Strengths:**

The manuscript is well-written and provides comprehensive details on all aspects of data collection and analysis. It presents significant work that is highly relevant to the journal.
The study offers several notable strengths, including its implications for estimating heart rate during sleep without relying on PPG sensors. It demonstrates that wrist-worn accelerometers can serve as a robust alternative for HR estimation, even in uncontrolled at-home settings where individuals may share their bed. This approach opens new avenues for analysis in studies that use accelerometers without HR signals. Overall, this work represents a valuable contribution to the field of HR estimation from wrist-worn accelerometer recordings.

**Summary Of The Paper:**

​​This study introduces the dataset of wrist-worn accelerometer recordings paired with electrocardiogram references in at-home settings, aimed at exploring IMU-only heart rate (HR) estimation via ballistocardiograms. The proposed frequency-based method effectively extracts HR while rejecting motion artifacts, achieving a mean absolute error of 0.88 bpm. These results highlight the potential of IMU-only HR estimation to uncover valuable health insights in large-scale studies.

**Weaknesses:**

While the work is innovative and relevant to the journal, I have the following concerns:

The manuscript mentions, “42 participants were recruited via convenience sampling in and around anonymized (18 female, 24 male, ages 20–36, M=26.48 years). Fifteen participants reported sharing their bed with someone else—none on a separate mattress.” It would be valuable for the authors to include an analysis focused on participants who slept alone (not necessarily divided by gender as mentioned in Table 2) to provide a more detailed understanding of the dataset and the results.

Additionally, including an analysis identifying the key factors contributing to low MAPE would strengthen the study.  Specifically, examining variables such as age groups, health conditions, and physical activity levels, in addition to gender, could offer deeper insights and enhance the robustness of the findings.

Clarification is also needed regarding the inclusion of participants with known sleep disorders, as these conditions could potentially affect heart rate during sleep. If applicable, this information should be discussed.

Furthermore, the manuscript would benefit from a comparison of the calculated heart rates with clinical resting heart rates, if such data were available for any participants. This comparison could provide further validation of the HR estimation method.

“From there until the last 60-second window when participants wore the movisens EcgMove 4 before their approximate wake time, we generally processed the synchronized raw data in 3-minute windows with a 2-minute overlap.” It would be beneficial for the authors to provide a detailed explanation for selecting this specific data processing approach and these particular metrics. Clarifying the rationale behind these choices would enhance the reader's understanding of the methodology and its implications for the study's findings.

The availability of the dataset for public access is not clear. Making the dataset publicly available would significantly contribute to the transparency and reproducibility of the research.

---

> ### Author Rebuttal · Authors · 2024-09-01
>
> 1. **Analysis of participants sleeping alone**\
> We have now added the following to Table II showing an 0.11 MAE increase.
>
> | alone | algorithm | MAE  | RSME | Cor  |
> | ----- | --------- | ---- | ---- | ---- |
> | Y     | Nightbeat | 0.83 | 2.23 | 0.82 |
> |       | Baseline  | 3.11 | 4.14 | 0.00 |
> | N     | Nightbeat | 0.94 | 2.15 | 0.82 |
> |       | Baseline  | 2.88 | 3.70 | 0.00 |
>
> 2. **Impact of demographics and other conditions on the MAE**\
> The MAE for female participants is slightly lower: 0.86 versus 0.87. Please find the correlations (incl. p-vals) with MAE below:
>
> | Variable    | Cor   | p-val |
> | ----------- | ----- | ----- |
> | Age         | 0.18  | 0.27  |
> | body weight | 0.03  | 0.84  |
> | body size   | -0.05 | 0.77  |
> | acc. mag. sleep | -0.20 | 0.24 |
>
> All correlations are on Nightbeat-DB since the AW dataset includes no demographics. We don't have information about health conditions nor activity during the day. We updated Section VI.
>
> 3. **Participants with sleep disorders**\
> None of the participants reported that they had a sleeping disorder as part of our questionnaire. We included this in our revision.
>
> 4. **Comparison to clinical resting heart rates**\
> While we cannot conduct this on Nightbeat-DB itself, performing such a comparison (e.g. on the UK Biobank) was one of the motivations for collecting Nightbeat-DB.
>
> 5. **Choices behind algorithm details: processing across 3 minutes**\
> In short, 20-second windows are not long enough to identify motion artifacts (i.e. abrupt wrist motions) or long-term HR trends (i.e., calculate the HR curve). Thus, we require longer processing windows even though we predict metrics much more granularly.
> Having analyzed our dataset, abrupt motion artifacts during sleep were rare and infrequent. Thus, motion artifact detection via outlier detection of the total magnitude of the STFT worked well. To ensure that each processing window is much longer than a potential motion artifact, we use a 3-minute processing window. 3-minute windows also proofed sufficient to identify meaningful HR curves.
> We revised Section III.D.
>
> 6. **availability of our dataset for public access**\
> Nightbeat-DB will be publicly available via a GitHub repository together with the entire code for processing Nightbeat and the full analysis of our manuscript. No further approval will be required for access. We will further apply for Nightbeat-DB to be included in PhysioNet (https://physionet.org/) to increase visibility & ease of access.

---

### Decision · Program_Chairs · 2024-09-23

Accept